# Impact of the Addition of Fruits of Kamchatka Berries (*L. caerulea* var. *kamtschatica*) and Haskap (*L. caerulea* var. *emphyllocalyx*) on the Physicochemical Properties, Polyphenolic Content, Antioxidant Activity and Sensory Evaluation Craft Wheat Beers

**DOI:** 10.3390/molecules28104011

**Published:** 2023-05-10

**Authors:** Justyna Belcar, Ireneusz Kapusta, Tomasz R. Sekutowski, Józef Gorzelany

**Affiliations:** 1Doctoral School, University of Rzeszów, St. Rejtana 16C, 35-959 Rzeszów, Poland; 2Department of Food and Agriculture Production Engineering, University of Rzeszów, St. Zelwerowicza 4, 35-601 Rzeszów, Poland; jgorzelany@ur.edu.pl; 3Department of Food Technology and Human Nutrition, University of Rzeszów, St. Zelwerowicza 4, 35-601 Rzeszów, Poland; ikapusta@ur.edu.pl; 4Institute of Soil Science and Plant Cultivation, National Research Institute, Puławy, Department of Weed Science and Tillage Systems, Orzechowa 61 St., 50-540 Wroclaw, Poland; t.sekutowski@iung.wroclaw.pl

**Keywords:** kamchatka berry, haskap, wheat beers, beer quality, bioactive compounds, antioxidant activity, polyphenol profile

## Abstract

Kamchatka berry (*Lonicera caerulea* var. *kamtschatica*) and haskap (*Lonicera caerulea* var. *emphyllocalyx*) fruit are important sources of bioactive compounds, mainly polyphenols, but also macro- and microelements. Physico-chemical analysis showed that wheat beers with added fruit were characterised by an average 14.06% higher ethanol content, lower bitterness and intense colour compared to the control, which was a wheat beer without added fruit. Wheat beers enriched with kamchatka berry fruit, including the “Aurora” variety, had the highest polyphenolic profile (e.g., chlorogenic acid content averaged 7.30 mg/L), and the antioxidant activity of fruit-enriched wheat beers determined by the DPPH method showed higher antioxidant activity of wheat beers enriched with kamchatka berry fruit, while those determined by the FRAP and ABTS methods showed higher antioxidant activity of wheat beers enriched with haskap fruit, including the “Willa” variety. Sensory evaluation of the beer product showed that wheat beers enriched with kamchatka berry fruits of the “Duet” variety and haskap fruits of the “Willa” variety were characterised by the most balanced taste and aroma. On the basis of the conducted research, it follows that both kamchatka berry fruits of the “Duet” and “Aurora” varieties and haskap fruit of the “Willa” variety can be used successfully in the production of fruity wheat beers.

## 1. Introduction

In recent years, consumers have been looking for new trends in beer styles, including increasing interest in beers other than those made from 100% barley malt and enriched with raw materials with high health-promoting effects such as fruit. Wheat beers are top-fermented products characterised by a fine, stable foam (thanks to a combination of two types of wheat and barley malt), little bitterness or haze intensity [1,2,3,4]. The palatability of wheat beers comes from the chemical compounds formed during fermentation, including phenols, aldehydes, esters and their derivatives [2,3,4]. Wheat beers are also characterised by a high content of antioxidant compounds, including polyphenols, melanoidins or vitamins [5].

Fruit enrichment of wheat beers can be used in the form of pulp, juice, concentrate or flavouring, which are most often introduced into beers during fermentation. The most common fruits used as a batch in beers are cherries, strawberries or exotic fruits, but there is also a search for new fruit species that could be used in the production of fruit beers, e.g., the fruit of the saskatoon berry or the defatted juice of the sea buckthorn [1,6]. The addition of fruit has a positive effect on both the health-promoting properties of the finished product (higher biological activity through the extraction of chemical compounds, e.g., polyphenols, from the added fruit into the beer) but also on the sensory properties (e.g., colour, aroma or taste of the beer product [7,8]). In global markets, one can find very popular radlers, i.e., a mix of beer with flavoured sugar syrup or fruit juice [6,9], as well as the characteristic Belgian beers of spontaneous fermentation—lambic with added raspberries “Framboise” or cherries “Kriek” [7,10,11].

Kamchatka berry (*Lonicera caerulea* var. *kamtschatica*) fruits are characterised by a blue–purple colouring of the skin, the shape resembles an ellipsoidal cylinder, the taste is juicy, sweet and sour with a waxy coating [12]. They are mainly cultivated in Japan, China, Russia (Siberia, Kamchatka) and northern European countries [13,14,15]. The chemical composition of kamchatka berry fruit varies depending on the cultivar, agrotechnical conditions and harvest date [13,14,16]. Kamchatka berry fruits contain sugars (mainly fructose and glucose) and minerals including K, P, Ca, Fe or organic acids (citric acid predominates, with malic and phytic acids also present in smaller amounts [13]). Kamchatka berry fruits are characterised by a high content of biologically active compounds with high antioxidant activity, including polyphenols, the main representatives of which are chlorogenic, coffeic and ferulic acids [12,13,17]. The most important compounds belonging to the flavonoid group found in kamchatka berry fruit are quercetin 3-*O*-rutinoside, quercetin 3-*O*-rhamnoglucoside, luteolin 7-*O*-rutinoside, luteolin 7-*O*-glucoside, diosmin, proanthocyanidins and catechins, while anthocyanins are mainly represented by cyanidin-3-*O*-glucoside [12,13,14,15,18]. Kamchatka berry fruits are also a good source of vitamin C [13,19]. Due to their rich chemical composition, kamchatka berries can be used for eye diseases (due to the anthocyanin content in the fruit), and also have antimicrobial, anticancer, detoxifying or anti-inflammatory properties [12,13,20]. In food processing, kamchatka berries are used for the production of jams, juices, tinctures, wines, frozen foods, as an addition to desserts, and can also be used as natural colourings [12,21].

In Japan, haskap (*Lonicera caerulea* var. *emphyllocalyx*) is also cultivated on the island of Hokkaido, which is becoming an increasingly popular plant worldwide, also in Poland, due to its fast harvest date (early June) but also due to its mechanical properties both during harvesting and during storage (more resistant to mechanical damage compared to kamchatka berries [22,23]). Similar to kamchatka berry fruits, haskap fruits have a purple skin colouration, but the shape is more like a barrel than a cylinder (length 1–2 cm, width 1 cm [22,23]. Haskap fruits have a rich chemical composition; they contain fibre, minerals including Ca and Mg, sugars (mainly glucose and fructose) and are a rich source of ascorbic acid. As with kamchatka berries, the main phenolic acids found in haskap fruit are chlorogenic acid, neochlorogenic acid and caffeic acid, and it is a source of anthocyanins, mainly cyanidin-3-*O*-glucoside. Haskap fruit is used in the cosmetic, pharmaceutical and food industries [24].

Due to their chemical properties (including their health-promoting properties), kamchatka berries and haskap fruits are used in many sectors of the food industry, while no studies have been found in the literature on the possibility of using these fruits as an additive for wheat beers. The aim of this study was to determine the quality of wheat beers enriched with kamchatka berry and haskap fruit and explore the practical applicability of the findings for expanding the range of fruit beers in the food industry. To achieve this, we analysed the physicochemical properties, polyphenolic content, and antioxidant activity of the resulting beers, as well as conducted sensory evaluations.

## 2. Results and Discussion

### 2.1. Physicochemical Characteristics of Wheat Beers

The results of the evaluation of the physico-chemical parameters of wheat beers enriched with kamchatka berry and haskap fruit are presented in Table 1.

The obtained wheat beers enriched with fruit were characterised by an average apparent extract of 3.15% *m/m*, while the real extract of the wheat beers was between 5.57 and 5.80% *m/m*, except for BW beer, whose real extract was significantly lower. The addition of fruit to wheat beers increased the original extract content by an average of 6.12% compared to the control beer (CB), and the addition of Kamchatka berry fruit had a greater effect on the original extract content compared to haskap fruit (Table 1). In a study by Gorzelany et al. [6], adding the fruits of the saskatoon berry to wheat beer, they obtained an average apparent extract of 4.44% *m/m*, an average real extract of 5.16% *m/m* and an average original extract of 13.30% *m/m*.

The lowest degree of final apparent attenuation was recorded for wheat beer without fruit addition (CB), while enrichment of the beer with kamchatka berry pulp increased the degree of final apparent attenuation by an average of 11.79%, and by an average of 7.52% when haskap fruit pulp was added. Wheat beer enriched with “Willa” haskap fruit pulp was characterised by a significantly higher degree of final real attenuation (Table 1). The addition of fruit to wheat beer increased the ethanol content by an average of 16.41% for kamchatka berry fruit and by an average of 11.39% for haskap fruit compared to the finished beer product constituting the control sample (CB; Table 1). Fruity wheat beers (enriched with saskatoon berry pulp) produced by Gorzelany et al. [6] were characterised by a lower ethanol content of 4.23% *v/v* on average. In cherry and blueberry fruit-enriched beers, Yang et al. [8] determined an alcohol content of 3.5% *v*/*v*. The ethanol content in raspberry beers was between 2.8–3.5% *v/v* [9]. In a study by Baigts-Allende et al. [7] beers with sour cherries were characterised by an alcohol content of 3.2–8.0% *v*/*v*, with raspberries 2.5–5.7% *v/v* and with blackcurrant 7.1% *v*/*v*, while Nedyalkov et al. [25] obtained barley beers with blueberry added with an ethanol content of 5.13% *v*/*v*. The energy value of the fruit wheat beers was between 50.53 and 61.13 kcal/100 mL and, with the exception of BW beer, was on average 8.86% higher than that of the control beer (CB; Table 1). The caloric content of wheat beers enriched with pulp from ozonated and non-ozonated fruits of the saskatoon berry was relatively lower and averaged 50.66 kcal/100 mL [6].

The colour of the wheat beers enriched with fruit was characterised by a purple hue, the intensity of which came from the fruit addition. On average, the colour of wheat beers enriched with haskap fruit was one unit lower than the average colour of wheat beers enriched with kamchatka berry fruit (Table 1; Figure 1). Wheat beers enriched with saskatoon berries were characterised by a colour intensity of 23.1–26.9 EBC units [6]. Beers with added blackcurrant were characterised by a colour intensity of 14.97 EBC units [7]. In a study by Patraşcu et al. [9] raspberry beers were characterised by a colour of 21.16 EBC units.

The acidity of wheat beers enriched with kamchatka berry fruit was on average 6.95% lower than that of beers enriched with haskap fruit irrespective of the variety used; moreover, the acidity of beers enriched with kamchatka berry fruit was on average 27.7% and that of beers with haskap fruit was on average 32.5% higher than that of the control beer without added fruit (CB; Table 1). The control wheat beer was characterised by the highest pH, while the addition of fruit lowered the parameter in question by an average of 15.1% (Table 1). Wheat beers with the addition of saskatoon berry were characterised by an average acidity of 3.55–4.22, while the pH was 4.40–4.47 [6]. In a study by Patraşcu et al. [9] raspberry beers were characterised by acidity and pH, respectively: 2.84–3.50 and 4.24. Nardini and Garaguso [10] analysing fruit beers obtained a pH in the range of 3.56–4.86. It is worth noting that a lower pH of the finished beer product represents a lower risk of infection by undesirable microflora of the finished beer product [26].

The perception of bitterness in wheat beers enriched with fruit was significantly lower than the control beer (CB); moreover, beers enriched with “Duet” kamchatka berry and “Lori” haskap fruit were characterised by 12.6% and 13.3% lower bitterness, respectively, compared to fruit-enriched beers enriched with “Aurora” and “Willa” fruit (Table 1). An important factor influencing the content of bittering compounds is, among other things, the cooking time of the wort with hops, on which the rate of the protein–polyphenol reaction depends [2,27]. The addition of kamchatka berry or haskap fruit pulp characterised by a sugar content of 1.5–2.1 g·100 g^−1^ and 1.2–2.1 g·100 g^−1^, respectively, also reduces the bitterness sensation in fruity wheat beers [28]. The carbon dioxide content of all wheat beers ranged from 0.44 to 0.48% (Table 1), a similar carbon dioxide content in fruit wheat beers was obtained by Gorzelany et al. [6].

### 2.2. Content of Bioactive Compounds in Fruit Wheat Beers

Among the many bioactive chemical compounds contained in beer, polyphenols, but also vitamins, melanoids or bitter acids are noteworthy [29,30]. Table 2 shows the antioxidant activity (determined by DPPH, FRAP and ABTS^+^ methods) of fruited wheat beers enriched with kamchatka berry and haskap fruit.

The antioxidant activity of wheat beers determined by three methods showed that fruit-enriched beers were characterised by higher activity compared to beer without added fruit (CB control; Table 2). The highest antioxidant activity determined by the DPPH method was characterised by wheat beers enriched with kamchatka berry fruits of the “Duet” variety, while the antioxidant activity of beers with the addition of haskap was on average 31.5% lower compared to beers with the addition of kamchatka berry fruits (DPPH method; Table 2). Wheat beers enriched with haskap fruit, on the other hand, were characterised by a significantly higher reducing capacity (FRAP method) by an average of 19.7% and antioxidant activity (ABTS method) by an average of 22.6% compared to wheat beers with kamchatka berry fruit (Table 2). Kamchatka berry fruits used in the production of fruit wheat beers were characterised by a reduced capacity of 30.52–37.67 μM Fe^2+^·g^−1^ f.w. and antioxidant activity (ABTS method) of 1.97–2.06 mM TE·100 g^−1^ f.w., while haskap fruit had a reducing capacity of 33.17–35.22 μM Fe^2+^·g^−1^ f.w. and antioxidant activity (ABTS method) of 2.12–2.21 mM TE·100 g^−1^ f.w. [28]. Wheat beers enriched with non-ozonated saskatoon berry had antioxidant activities of 2.68 mM TE/L (DPPH method) and 2.20 mM TE/L (ABTS method) and a reducing capacity of 1.80 mM Fe^2+^/L [6].

The polyphenolic compounds found in beers are mainly derived from the malt used (70–80%) and the hops [11]. The degree of fineness of the malt, as well as the conditions of the mashing and boiling process with hops, significantly affect the total polyphenol content [27]. Polyphenolic compounds are diverse substances with various biologically active effects, including antioxidant and antiradical activity [31]. Enrichment of wheat beers with fruit had a significantly positive effect on the content of total polyphenols in the finished product in that significant differences were observed between added fruits. Beers enriched with haskap fruit were characterised by an average 31.6% higher content of total polyphenols compared to beers with kamchatka berry fruit added, irrespective of variety (Table 3). Crushing the fruit and feeding it to the fermenting wort in the form of pulp influences the extent to which phenolic compounds go into the finished product by increasing the contact of the solution, which leaches and transfers the chemical compounds found in the fruit through the ruptured cell wall, thus enriching the finished beer product [26]. Wheat beers enriched with saskatoon berries were characterised by total polyphenol content in the range of 377–413 mg GAE/L [6]. The content of total polyphenols in beers enriched with cornelian cherry fruit was 350 mg GAE/L [32], and with goji berries was 415 mg GAE/L [33].

Identification of polyphenolic compounds in wheat beers enriched with kamchatka berry and haskap fruit was made on the basis of analysis of characteristic spectral data. Nine polyphenolic compounds were identified, the spectral properties of which are shown in Table 3. UPLC-PDA-MS-MS analysis showed that six polyphenolic compounds belonging to the flavonol group (compounds **4**; **6–9**) were identified in wheat beers without added fruit, of which the representative compounds were kempferol and quercetin derivatives, of which the highest concentration was dihydroquercetin 3-*O*-glucoside (0.98 mg/L; Table 3) and one chemical compound belonging to the phenolic acid group (caffeoylglucaric acid; Table 3). Nine chemical compounds belonging to the group of hydroxycinnamic acid derivatives (compounds **1–3**; **5**) and to the group of flavonols (compounds **4**; **6–9**) were identified in wheat beers enriched with kamchatka berries. Chlorogenic acid had the highest concentration among the analysed compounds (average 6.58 mg/L), as well as, although at a much lower concentration, quercetin-3-*O*-rutinoside (average 1.66 mg/L; Table 3). Nine compounds belonging to the group of hydroxycinnamic acid derivatives (compounds **1–3**; **5**) and to the group of flavonols (compounds **4**; **6–9**) were identified in wheat beers with the addition of haskap fruit for the “Willa” variety, while quercetin-3-*O*-rutinoside and quercetin-3-*O*-glucoside were not detected for the wheat beer with addition on “Lori”; variety, and chlorogenic acid had the highest concentration (average 2.51 mg/L; Table 3).

The lowest content of polyphenolic compounds identified in the wheat beers enriched with fruit, 16.72% higher than the control beer (CB), was characterised by the beer product with the addition of “Lori” haskap fruit, while the wheat beer enriched with “Willa” haskap fruit was characterised by an average 63.12% higher content of identified polyphenolic compounds compared to BL beer (Table 3). Wheat beers enriched with kamchatka berry fruit were characterised by a significantly higher content of identified polyphenolic compounds (by 30.79% on average compared to BW beer; Table 3). The content of identified polyphenolic compounds in wheat beers with the addition of non-ozonated saskatoon berries averaged 8.87 mg/L [6].

Of the phenolic acids found in fruit-enriched wheat beers, the content of chlorogenic acid was the highest regardless of fruit addition, with beers enriched with kamchatka berry fruit having a significantly higher content (by 61.85% on average; Table 3). In wheat beers enriched with non-ozonated saskatoon berry, the average chlorogenic acid content was 1.82 mg/L, while neochlorogenic acid content was 1.14 mg/L [6]. In barley beers with blueberry (167 g/L fruit addition), the content of chlorogenic acid—90.19 mg/L and neochlorogenic acid—52.24 mg/L [25]. Phenolic acids, including chlorogenic acid and neochlorogenic acid, are characterised by antioxidant activity, inhibit damage to DNA structure and influence fruit aroma [34].

Kaempferol and quercetin glycosides have also been identified in fruit wheat beers, which have strong anticancer, antioxidant or supportive properties in cardiovascular disease [35]. The average content of quercetin derivatives in beers enriched with kamchatka berries was 3.57 mg/L, while wheat beers enriched with haskap fruit and the control beer (CB) were characterised by a lower content of quercetin derivatives by an average of 53.78% and 58.26%, respectively, compared to beers with kamchatka berries (Table 3). As reported by numerous researchers [36,37,38,39], the average quercetin content in barley beers ranges from 0.06 to 1.79 mg/L. Flavone glycosides as well as chlorogenic acid impart an astringent and acidic mouthfeel and, although to a much lower degree, also a bitter sensation, which affects the sensory experience of the finished beer product [35].

### 2.3. Sensory Analysis of Fruity Wheat Beers

Not only the acceptability but also the attractiveness of the finished fruit beer product are important characteristics that influence consumer choice. The taste and aroma qualities of fruity wheat beers can determine whether consumers will decide to purchase a particular type of beer again or whether this purchase will be a one-off. The results of the sensory evaluation of the fruit wheat beers carried out by the 10-person panel are shown in Table 4 and Figure 2.

The addition of kamchatka berry and haskap fruit to wheat beer, especially of the “Willa” variety, resulted in positive flavour and aroma qualities of the finished beer product (Table 4). The taste and aroma of beer are influenced not only by the raw materials used but also by the products of the fermentation process (including aldehydes, phenols or esters) affecting the taste profile of a given beer. Significant quality attributes of fruity wheat beers with kamchatka berry and haskap fruit were malty, fruity, full, intense and fresh taste and aroma (Figure 2). In a study on the possibility of enriching wheat beers with the fruit of the saskatoon berry, Gorzelany et al. [6] obtained similar results for the taste and aroma profile, and a sour aftertaste was also clearly perceptible in the finished product, irrespective of the ozonation process applied to the fruit. Chemical compounds that are important for the flavour of beer are formed by interactions between carbonyl compounds, esters, sulphur compounds, alcohols, phenolic compounds or organic acids [40]. Beers characterised by fruity notes, with a sweet aftertaste and pleasant aroma are more preferred and desired by consumers over traditional types of beers [41,42].

Wheat beers enriched with “Aurora” kamchatka berry fruit and “Lori” haskap were characterised by the most stable beer foam and carbon dioxide saturation. By far the highest bitterness sensation among the obtained fruity wheat beers was characterised by the finished beer product enriched with kamchatka berry fruits of the “Aurora” variety, a result different from the results of the physico-chemical analysis of the analysed beer (Table 1; Table 4). The perception of a sour or bitter fruity wheat beer is related to the varying content of the polyphenols responsible, including chlorogenic acid [27]. This section may be divided by subheadings. It should provide a concise and precise description of the experimental results, their interpretation, as well as the experimental conclusions that can be drawn.

## 3. Materials and Methods

### 3.1. Material

The research material consisted of fruit of two varieties of kamchatka berry: “Duet” and “Aurora” from a nursery cultivation located in Tyczyn (49°57′52″ N 22°2′47″ E, Subcarpathian Province, Poland) in 2022 and two haskap cultivars: “Lori” and “Willa”, which were obtained from “Korfanty” (49°41′41″ N 22°5′3″ E, Grabownica Starzeńska, Subcarpathian Voivodeship, Poland) in 2022. The results of physico-chemical analyses of the fruit are described in Gorzelany et al. [28].

Grains of winter common wheat of the “Elixer” variety from a field experiment conducted in 2021 in Przeworsk (50°03′31″ N 22°29′37″), Subcarpathian Province, Poland, were used to produce wheat beers. A 5-day wheat malt was prepared from the grain (the methodology of the malting process is described in Belcar et al. [43]). The wheat malt (“Elixer” variety) had the following characteristics: extract potential—84.8% d.m., total protein content—11.0% d.m., content of soluble protein—4.51% d.m., diastatic power—337 WK, and degree of final attenuation—80.9%.

The raw material charge for brewing the wheat beers consisted of 60% commercial grist barley malt and 40% grist wheat malt. Commercial barley malt from the Viking Malt malting plant in Strzegom, Poland, was used for brewing. The barley malt had the following characteristics: extract potential—80.0% d.m., total protein content—11.4% d.m., content of soluble protein—3.75% d.m., diastatic power—324 WK, and degree of final attenuation—82.1%.

### 3.2. Production of Beers

The production process was carried out using the infusion method in the laboratory of the Department of Agricultural and Food Production Engineering at the University of Rzeszów. Ground barley and wheat malts weighing 4.0 kg (2.4 kg of barley malt and 1.6 kg of wheat malt) were placed in a ROYAL RCBM-40N mash kettle (Expondo; Poland; assuming a process efficiency of 80%) and 12.0 l of water was added (3 l of water for each kilogram of malt). The mashing, boiling process with hops and cooling of the beer wort were carried out according to the methodology described in Gorzelany et al. [6].

The extract of the produced beer worts was 12.0°P. After cooling, the wort was transferred to fermentation containers of 30 L each and inoculated with *Saccharomyces cerevisae* Fermentis Safale US-05 yeast (6 × 10^9^/g), which had previously undergone a rehydration process according to the manufacturer’s instructions (0.58 g d.m./L of wort). The fermentation process was carried out for 21 days at 21 °C. After 7 days of fermentation, 5% d.m./L of wort was added fruits to the fermenting beer) and left to ferment for another 14 days. Once fermentation was complete, the green beer was bottled, with a solution of sucrose (0.3%) in water added beforehand for refermentation and to achieve adequate beer saturation. The resulting beers were kept at 20 °C. Sensory and physico-chemical tests were performed one month after bottling.

Wheat beers obtained from wheat malt without fruit addition were designated CB (control beer), with the addition of kamchatka berry fruits of the “Duet” variety designated BD, while those of the “Aurora;” variety was designated BA. Wheat beers enriched with haskap fruit of the “Lori” variety were designated BL, and with the addition of the “Willa” variety were designated BW. A total of five variants of wheat beers were produced.

### 3.3. Analysis of Quality Indicators for Beers

The alcohol content, apparent extract, real extract and original extract of the beer, degree of final apparent and real attenuation, total acidity, pH, colour, carbon dioxide content, bitterness content and energy value of the beer were determined according to the methodology given in Belcar et al. [44]. The analyses were performed in three replications.

### 3.4. Content of Bioactive Compounds in Fruit Beers

The total polyphenol content by the Folin–Ciocalteu method and the polyphenol profile in the analysed beers were determined according to the methodology given by Gorzelany et al. [6]. The analyses were performed in three replications.

### 3.5. Antioxidant Activity

The antioxidant activity of the fruit beers (by DPPH, FRAP and ABTS methods) was determined according to the methodology given by Gorzelany et al. [6]. The analyses were performed in three replications.

### 3.6. Sensory Analysis

Sensory analysis was performed by an expert team of ten assessors (five women and five men, aged 30–40 years) in the sensory evaluation laboratory according to EBC method 13.13 [45]. Beer samples were served after cooling to 10 °C coded in random order in 250 mL transparent plastic cups. Water was administered between each evaluation. Sensory analysis of the beers was performed according to the methodology given by Belcar and Gorzelany [1]. The sensory profile of fruit beers made with kamchatka berry and haskap fruit was compared with the control beer (without added fruit).

### 3.7. Statistical Analysis

The results of the analyses of the fruit beers are presented as mean values with standard deviation. Statistical analysis of the results was performed using Statistica 13.3 statistical software (TIBCO Software Inc., Tulsa, OK, USA). Analyses used a two-factor ANOVA analysis of variance in a complete randomised design with a significance level of α = 0.05 for the individual results of the physico-chemical analysis, polyphenol content, and antioxidant activity of the fruit beers. Comparisons of mean values were made using the HSD Tukey test.

## 4. Conclusions

Fruity wheat beers enriched with kamchatka berry and haskap fruit were characterised by higher colour intensity and lower bitterness and higher ethanol content compared to the control. The antioxidant activity of wheat beers varied; beers enriched with kamchatka berry fruit were characterised by higher antioxidant activity determined by the DPPH method, while beers with the addition of haskap fruit, including the “Willa” variety, were characterised by the other methods determined. Analysis of the polyphenolic profile showed a significantly higher content of polyphenolic compounds in wheat beers enriched with kamchatka berry fruit. Sensory evaluation of the beer product showed that wheat beers with the addition of haskap fruit of the “Willa” variety and kamchatka berry fruit of the “Duet” variety were characterised by the most balanced taste and aroma. On the basis of the conducted research, it follows that both the kamchatka berry fruits of the “Duet” and “Aurora” varieties and the haskap variety “Willa” can be successfully used in the production of fruity wheat beers.

## Figures and Tables

**Figure 1 molecules-28-04011-f001:**
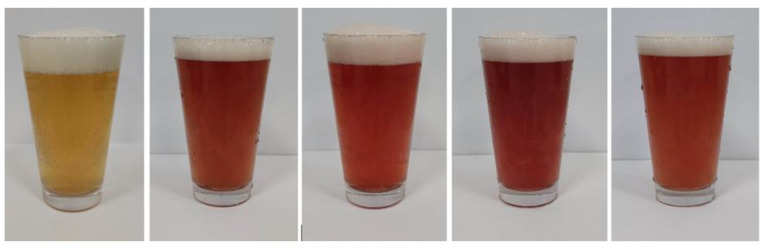
Appearance of the wheat beers obtained—from left: control (CB), wheat beer with kamchatka berry “Duet” (BD) and “Aurora” (BA), and with haskap fruits “Lori” (BL) and “Willa” (BW).

**Figure 2 molecules-28-04011-f002:**
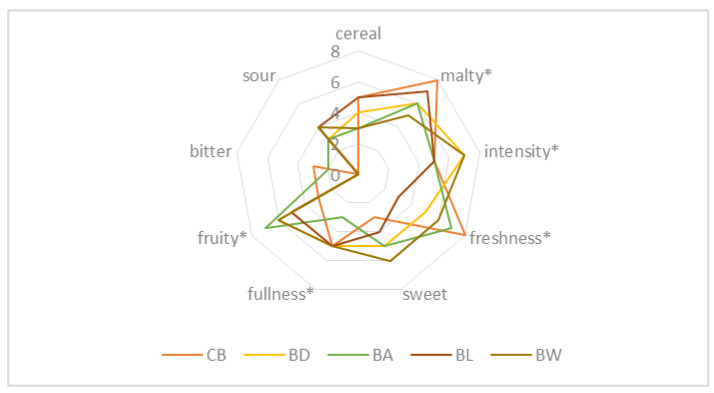
Sensory profile of wheat beers—control (CB) and with the addition of kamchatka berry fruits of the “Duet” variety (BD) and the “Aurora” variety (BA) and enriched with haskap fruits of the “Lori” variety and the “Willa” variety. (* indicates attributes which were statistically different at *p* < 0.05).

**Table 1 molecules-28-04011-t001:** Results of physico-chemical analysis of wheat beers enriched with kamchatka berry and haskap fruit.

Parameter	CB	BD	BA	BL	BW
Apparent extract (%; *m/m*)	4.03 ^d^ ± 0.03	2.58 ^a^ ± 0.04	3.46 ^c^ ± 0.04	3.52 ^c^ ± 0.08	3.05 ^b^ ± 0.05
Real extract (%; *m/m*)	5.60 ^b^ ± 0.10	5.57 ^b^ ± 0.03	5.80 ^c^ ± 0.10	5.63 ^b^ ± 0.03	3.86 ^a^ ± 0.01
Original extract (%; *m/m*)	13.81 ^b^ ± 0.01	15.96 ^e^ ± 0.04	14.88 ^d^ ± 0.07	14.57 ^c^ ± 0.06	13.44 ^a^ ± 0.04
Degree of final apparent attenuation (%)	70.82 ^a^ ± 0.10	83.83 ^e^ ± 0.03	76.75 ^c^ ± 0.05	75.84 ^b^ ± 0.04	77.31 ^d^ ± 0.01
Degree of final real attenuation (%)	59.45 ^a^ ± 0.05	65.10 ^d^ ± 0.10	61.02 ^b^ ± 0.08	61.36 ^c^ ± 0.06	71.28 ^e^ ± 0.05
Content of alcohol (%; *m/m*)	4.28 ^a^ ± 0.04	5.48 ^e^ ± 0.03	4.76 ^c^ ± 0.04	4.68 ^b^ ± 0.05	4.98 ^d^ ± 0.04
Content of alcohol (%; *v*/*v*)	3.40 ^a^ ± 0.10	4.36 ^d^ ± 0.04	3.79 ^b^ ± 0.01	3.72 ^b^ ± 0.08	3.96 ^c^ ± 0.04
Colour (EBC units)	22.4 ^a^ ± 0.4	29.7 ^c^ ± 0.3	31.5 ^e^ ± 0.5	28.4 ^b^ ± 0.2	30.5 ^d^ ± 0.4
Titratable acidity (0.1 M NaOH/100 cm^3^)	2.71 ^a^ ± 0.01	3.33 ^b^ ± 0.03	4.18 ^d^ ± 0.04	4.10 ^d^ ± 0.10	3.95 ^c^ ± 0.05
pH	4.83 ^c^ ± 0.02	4.19 ^b^ ± 0.07	4.01 ^a^ ± 0.06	4.04 ^a^ ± 0.04	4.16 ^b^ ± 0.04
Bitter substances (IBU)	15.4 ^e^ ± 0.2	11.1 ^a^ ± 0.1	12.7 ^c^ ± 0.3	11.7 ^b^ ± 0.4	13.5 ^d^ ± 0.5
Content of carbon dioxide (%)	0.46 ^a^ ± 0.04	0.47 ^a^ ± 0.02	0.44 ^a^ ± 0.04	0.47 ^a^ ± 0.02	0.48 ^a^ ± 0.02
Energy value (kcal/100 mL)	52.85 ^b^ ± 0.05	61.13 ^e^ ± 0.13	57.05 ^d^ ± 0.05	55.78 ^c^ ± 0.05	50.53 ^a^ ± 0.07

Data are expressed as a mean values (*n* = 3) ± SD; SD—standard deviation. Mean values within rows with different letters are significantly different (*p* < 0.05). CB—wheat beer without added fruit; BD—wheat beer with added kamchatka berry “Duet”; BA—wheat beer with added kamchatka berry “Aurora”; BL—wheat beer with added haskap “Lori”; BW—wheat beer with added haskap “Willa”.

**Table 2 molecules-28-04011-t002:** Antioxidant activity of fruited wheat beers.

Antioxidant Assay	CB	BD	BA	BL	BW
DPPH [mM TE/L]	1.04 ^a^ ± 0.06	2.14 ^e^ ± 0.06	1.58 ^d^ ± 0.02	1.36 ^c^ ± 0.04	1.19 ^b^ ± 0.06
FRAP [mM Fe^2+^/L]	0.86 ^a^ ± 0.04	2.07 ^b^ ± 0.03	2.09 ^b^ ± 0.05	2.47 ^c^ ± 0.03	2.71 ^d^ ± 0.04
ABTS^+^ [mM TE/L]	1.01 ^a^ ± 0.04	1.37 ^c^ ± 0.03	1.27 ^b^ ± 0.06	1.97 ^d^ ± 0.03	1.44 ^c^ ± 0.04

Data are expressed as a mean values (*n* = 3) ± SD; SD—standard deviation. Mean values within rows with different letters are significantly different (*p* < 0.05). CB—wheat beer without added fruit; BD—wheat beer with added kamchatka berry “Duet”; BA—wheat beer with added kamchatka berry “Aurora”; BL—wheat beer with added haskap “Lori”; BW—wheat beer with added haskap “Willa”.

**Table 3 molecules-28-04011-t003:** Polyphenol content and polyphenol profile identified by UPLC-PDA-MS-MS.

					CB	BD	BA	BL	BW
Total polyphenols content (mg GAE/L)					134.0 ^a^ ± 0.2	180.3 ^b^ ± 0.3	181.2 ^b^ ± 0.4	276.3 ^d^ ± 0.3	252.3 ^c^ ± 0.7
Compound (mg/L)	Rt (min)	λ_max_ (nm)	(M-H) *m*/*z*						
			MS	MS/MS					
Neo-chlorogenic acid	2.15	288 sh, 324	353	191	<LOQ	0.82 ^c^ ± 0.05	0.68 ^b^ ± 0.01	0.48 ^a^ ± 0.02	0.67 ^b^ ± 0.03
Chlorogenic acid	2.70	299 sh, 327	353	191	<LOQ	5.86 ^c^ ± 0.03	7.30 ^d^ ± 0.01	1.20 ^a^ ± 0.02	3.81 ^b^ ± 0.02
Unspecified caffeic acid derivative	3.02	298 sh, 322	507	353, 161	<LOQ	0.25 ^b^ ± 0.08	0.26 ^b^ ± 0.09	0.17 ^a^ ± 0.02	0.53 ^c^ ± 0.08
K-3-*O-p*-hydroxybenzoyl-glc	3.56	264, 347	567	447, 285	0.65 ^d^ ± 0.01	0.44 ^b^ ± 0.09	0.32 ^a^ ± 0.01	0.30 ^a^ ± 0.01	0.49 ^c^ ± 0.02
Caffeoylglucaric acid	3.61	299 sh, 327	371	179	0.41 ^c^ ± 0.01	0.38 ^b^ ± 0.00	0.36 ^b^ ± 0.00	0.29 ^a^ ± 0.00	0.59 ^d^ ± 0.02
Dihydroquercetin 3-*O*-glc	3.73	255, 354	465	303	0.98 ^d^ ± 0.02	0.80 ^c^ ± 0.02	0.42 ^a^ ± 0.02	0.41 ^a^ ± 0.02	0.70 ^b^ ± 0.03
Q-3-*O*-glc-pent	3.88	253, 354	595	301	0.17 ^a^ ± 0.00	0.86 ^d^ ± 0.04	0.74 ^c^ ± 0.00	0.71 ^c^ ± 0.00	0.53 ^b^ ± 0.02
Q-3-*O*-rut	4.16	255, 355	609	301	0.06 ^a^ ± 0.00	0.92 ^c^ ± 0.02	2.39 ^d^ ± 0.07	<LOQ	0.73 ^b^ ± 0.00
Q-3-*O*-glc	4.35	253, 352	463	301	0.28 ^c^ ± 0.00	0.86 ^d^ ± 0.03	0.15 ^a^ ± 0.01	<LOQ	0.21 ^b^ ± 0.00
Total					2.54 ^a^ ± 0.05	11.19 ^d^ ± 0.32	12,70 ^e^ ± 0.17	3.05 ^b^ ± 0.46	8.27 ^c^ ± 0.22

Data are expressed as a mean values (*n* = 3) ± SD; SD—standard deviation. Mean values within a row with different letters are significantly different (*p* < 0.05). CB—wheat beer without added fruit; BD—wheat beer with added kamchatka berry “Duet”; BA—wheat beer with added kamchatka berry “Aurora”; BL—wheat beer with added haskap “Lori”; BW—wheat beer with added haskap “Willa”. <LOQ—below limit of quantification; K—kaempferol; Q—quercetin; glc—glucoside; rut—rutinoside; pent—pentoside.

**Table 4 molecules-28-04011-t004:** Sensory analysis of fruit-enriched wheat beers.

	CB	BD	BA	BL	BW
Aroma	3.51 ^a^ ± 0.41	4.13 ^ab^ ± 0.25	4.20 ^b^ ± 0.60	3.96 ^a^ ± 0.57	4.52 ^b^ ± 0.68
Taste	3.58 ^a^ ± 0.24	4.27 ^b^ ± 0.34	3.64 ^a^ ± 0.28	3.62 ^a^ ± 0.44	4.31 ^b^ ± 0.52
Foam stability	3.42 ^a^ ± 0.13	3.82 ^a^ ± 0.30	3.94 ^a^ ± 0.17	4.14 ^b^ ± 0.33	3.71 ^a^ ± 0.26
Bitterness	3.75 ^a^ ± 0.17	3.51 ^a^ ± 0.14	4.49 ^b^ ± 0.25	3.77 ^a^ ± 0.37	4.05 ^ab^ ± 0.34
Saturation	3.96 ^a^ ± 0.34	3.55 ^a^ ± 0.17	4.13 ^ab^ ± 0.27	4.28 ^b^ ± 0.29	3.68 ^a^ ± 0.33
Overall impression	3.56 ^a^ ± 0.12	3.97 ^b^ ± 0.34	3.94 ^b^ ± 0.50	3.81 ^ab^ ± 0.27	4.14 ^b^ ± 0.52

Data are expressed as a mean values (*n* = 10) ± SD; SD—standard deviation. Mean values within a row with different letters are significantly different (*p* < 0.05). CB—wheat beer without added fruit; BD—wheat beer with added kamchatka berry “Duet”; BA—wheat beer with added kamchatka berry “Aurora”; BL—wheat beer with added haskap “Lori”; BW—wheat beer with added haskap “Willa”.

## Data Availability

Not applicable.

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
