# Peer review of "Impact of the Addition of Fruits of Kamchatka Berries (L. caerulea var. kamtschatica) and Haskap (L. caerulea var. emphyllocalyx) on the Physicochemical Properties, Polyphenolic Content, Antioxidant Activity and Sensory Evaluation Craft Wheat Beers"

_molecules, 2023, doi:10.3390/molecules28104011_

Round 1

Reviewer 1 Report

Manuscript ID: molecules-2394864

This research investigated using Kamchatka berry and Haskap fruit to produce fruity wheat beers. The fruits are rich in bioactive compounds, and the resulting beers had a higher ethanol content, lower bitterness, and intense color compared to the control. Kamchatka berry-enriched beers had higher polyphenolic content, while Haskap-enriched beers had higher antioxidant activity. Sensory evaluation revealed the most balanced taste and aroma in beers enriched with Kamchatka berry fruits of the 'Duet' variety and Haskap fruits of the 'Willa' variety. The study concludes that both Kamchatka berry and Haskap fruit can be effectively used in the production of fruity wheat beers.

The manuscript appears to be novel as it explores the use of Kamchatka berry and Haskap fruit in the production of fruity wheat beers. The study also analyzes the physicochemical properties, polyphenolic content, and antioxidant activity of the resulting beers, as well as conducting sensory evaluations. Overall, the manuscript provides new insights into the use of these fruits in beer production and their potential benefits.

1. In my opinion, the last paragraph of the introduction should provide more information about what the authors did. For instance, “The aim of this study was to determine the quality of wheat beers enriched with Kamchatka berry and Haskap fruit and explore the practical applicability of the findings for expanding the range of fruit beers in the food industry. To achieve this, we analyzed the physicochemical properties, polyphenolic content, and antioxidant activity of the resulting beers, as well as conducted sensory evaluations”.

2. Table 1, please add the term 'Parameter' in the first column to represent the physicochemical data.

3.  In Table 2. Please add “Antioxidant assay” into the first column to indicate the antioxidant testing methods of DPPH, FRAP and ABTS assays.

4. In 3.6. Sensory analysis: Lines 373-374; An expert team of women and men, aged 30 - 40 years was used for sensory analysis. In my opinion, the narrow age range of participants recruited for the sensory analysis raises questions about the reason for this choice. Was it because of the target audience for this type of beer, or was it not really a choice? If it was not a deliberate choice, could this potentially introduce bias in the sensory study with regards to the general population who are likely to consume this type of beer? Please explain.

Minor editing of the English language is required. There are some typographical errors. 

Author Response

The authors are grateful for the contribution of the Reviewer.

1. In my opinion, the last paragraph of the introduction should provide more information about what the authors did. For instance, “The aim of this study was to determine the quality of wheat beers enriched with Kamchatka berry and Haskap fruit and explore the practical applicability of the findings for expanding the range of fruit beers in the food industry. To achieve this, we analyzed the physicochemical properties, polyphenolic content, and antioxidant activity of the resulting beers, as well as conducted sensory evaluations”.

Answer:

It has been corrected as suggested by the Reviewer (lines 92-96).

2. Table 1, please add the term 'Parameter' in the first column to represent the physicochemical data.

3.  In Table 2. Please add “Antioxidant assay” into the first column to indicate the antioxidant testing methods of DPPH, FRAP and ABTS assays.

Answer:

It has been corrected.

4. In 3.6. Sensory analysis: Lines 373-374; An expert team of women and men, aged 30 - 40 years was used for sensory analysis. In my opinion, the narrow age range of participants recruited for the sensory analysis raises questions about the reason for this choice. Was it because of the target audience for this type of beer, or was it not really a choice? If it was not a deliberate choice, could this potentially introduce bias in the sensory study with regards to the general population who are likely to consume this type of beer? Please explain.

Answer:

From our own research prior to the study on the effects of different types of additives (e.g. sea buckthorn juice; Belcar and Gorzelany 2022, lemongrass; Belcar and Gorzelany 2022, saskatoon berry; Belcar et. al. 2022), it was apparent that the group of people aged 30-40 years is the most interested in expanding the range of fruity wheat beers and the sensory research is therefore targeted at this group of people.

According to the Reviewer's comments, the manuscript has been revised by a native speaker.

Reviewer 2 Report

The submitted paper is quite interesting, well-designed and the results are presented and discussed in a clear way. However, I believe that the soundness of the paper is lower than its content. I have some suggestions for improvement.

The title of the paper must be adjusted to describe the used analyses for sample characterisation. 

l37 - chabge to health-promoting effects

l37 - 38: wheat beers produced solely from wheat barley do not have stable foam, and this is the main reason why commercial wheat beers are produced from mixtures of wheat and barley malt. This aspect must be stated.

In general, in the introduction part, the gap of the study is not mentioned. Please mention it. 

l87 - 89: Clearly define how was the quality of the beers determined. The word quality is too general.

L333: Chanhe he to The

L373: assessors instead of people

l376: what is oral water???

is ok, some modifiacations should be done

Author Response

The authors are grateful for the contribution of the Reviewer.

The title of the paper must be adjusted to describe the used analyses for sample characterisation. 

Answer:

The title has been corrected on 'Impact of the additive of fruits of kamchatka berries (L. caerulea var. kamtschatica) and haskap (L. caerulea var. emphyllocalyx) on the quality and pro-healthy profile craft wheat beers'

l37 - change to health-promoting effects - It has been corrected.

l37 - 38: wheat beers produced solely from wheat barley do not have stable foam, and this is the main reason why commercial wheat beers are produced from mixtures of wheat and barley malt. This aspect must be stated.

Answer:

It has been corrected. Lines 38-40: 'Wheat beers are top-fermented products characterised by a fine, stable foam (thanks to a combination of two types of wheat and barley malt), little bitterness or haze intensity [1-4].'

In general, in the introduction part, the gap of the study is not mentioned. Please mention it. 

l87 - 89: Clearly define how was the quality of the beers determined. The word quality is too general.

Answer:

It has been corrected. Lines 89 - 96: 'Due to their chemical properties (including their health-promoting properties), kamchatka berries and haskap fruits are used in many sectors of the food industry, while no studies have been found in the literature on the possibility of using these fruits as an additive for wheat beers. The aim of this study was to determine the quality of wheat beers enriched with kamchatka berry and haskap fruit and explore the practical applicability of the findings for expanding the range of fruit beers in the food industry. To achieve this, we analyzed the physicochemical properties, polyphenolic content, and antioxidant activity of the resulting beers, as well as conducted sensory evaluations.'

L333: Chanhe he to The

L373: assessors instead of people

Answer: It has been corrected.

l376: what is oral water???

Answer:

Oral water is a mistake. Should be water.

According to the reviewer's comments, the manuscript has been revised by a native speaker.

Round 2

Reviewer 2 Report

Dear authors,

I am very happy to see that you have modified the manuscript according to reviewers suggestions.

However, the modified title of the manuscript is not fine. It should be corrected, by changing the word additive by addition and the used anakyses should be clearly mentioned. As you have in the last sentence of the introduction where the gap and aim of the work is discussed.

After this minor modification the paper can be zccepted.

Author Response

Our team thanks Receiver for suggestion. The Title has been corrected.